# A Concise Review on Dysregulation of LINC00665 in Cancers

**DOI:** 10.3390/cells11223575

**Published:** 2022-11-11

**Authors:** Soudeh Ghafouri-Fard, Tayyebeh Khoshbakht, Bashdar Mahmud Hussen, Aria Baniahmad, Mohammad Taheri, Mohammadreza Hajiesmaeili

**Affiliations:** 1Department of Medical Genetics, School of Medicine, Shahid Beheshti University of Medical Sciences, Tehran 19835-35511, Iran; 2Men’s Health and Reproductive Health Research Center, Shahid Beheshti University of Medical Sciences, Tehran 16666-63111, Iran; 3Department of Pharmacognosy, College of Pharmacy, Hawler Medical University, Erbil 44001, Iraq; 4Center of Research and Strategic Studies, Lebanese French University, Erbil 44001, Iraq; 5Institute of Human Genetics, Jena University Hospital, 07743 Jena, Germany; 6Urology and Nephrology Research Center, Shahid Beheshti University of Medical Sciences, Tehran 16666-63111, Iran; 7Critical Care Quality Improvement Research Center, Loghman Hakim Hospital, Shahid Beheshti University of Medical Sciences, Tehran 16666-63111, Iran

**Keywords:** LINC00665, cancer, biomarker

## Abstract

Long Intergenic Non-Protein Coding RNA 665 (LINC00665) is an RNA gene located on the minus strand of chromosome 19. This lncRNA acts as a competing endogenous RNA for miR-4458, miR-379-5p, miR-551b-5p, miR-3619-5p, miR-424-5p, miR-9-5p, miR-214-3p, miR-126-5p, miR-149-3p, miR-379-5p, miR-665, miR-34a-5p, miR-186-5p, miR-138-5p, miR-181c-5p, miR-98, miR-195-5p, miR-224-5p, miR-3619, miR-708, miR-101, miR-1224-5p, miR-34a-5p, and miR-142-5p. Via influencing expression of these miRNAs, it can enhance expression of a number of oncogenes. Moreover, LINC00665 can influence activity of Wnt/β-Catenin, TGF-β, MAPK1, NF-κB, ERK, and PI3K/AKT signaling. Function of this lncRNA has been assessed through gain-of-function tests and/or loss-of-function studies. Furthermore, diverse research groups have evaluated its expression levels in tissue samples using microarray and RT-qPCR techniques. In this manuscript, we have summarized the results of these studies and categorized them in three sections, i.e., cell line studies, animal studies, and investigations in clinical samples.

## 1. Introduction

Long non-coding RNAs (lncRNAs) are a group of transcripts participating in the chromatin remodeling, transcriptional regulation of gene expression as well as post-transcriptional events, via different chromatin-associated routes and through interaction with other RNA molecules [1]. Their functions largely depend on their subcellular localization and interplays with DNA, RNA, and proteins [2]. In addition to their role in the modulation of chromatin function, they can influence the stability of mRNAs and their translation in the cytoplasm [2]. Their involvement in diverse biological and physiopathological functions has potentiated them as important participants in the pathoetiology of different disorders.

Long Intergenic Non-Protein Coding RNA 665 (LINC00665) is an RNA gene located on chr19:36,259,540-36,332,581. At least 33 splice variants have been identified for this gene (http://asia.ensembl.org/Homo_sapiens/Gene, accessed on 10 October 2022). This lncRNA has been shown to act as an oncogene in diverse malignancies. Function of this lncRNA has been assessed through gain-of-function tests and/or loss-of-function studies. Moreover, different groups have evaluated its expression levels in tissue samples using microarray and real time-quantitative polymerase chain reaction (RT-qPCR) techniques. Similar to some other lncRNAs, LINC00665 has been shown to encode a micropeptide. This micropeptide is called CIP2A-BP. Experiments in breast cancer cells have revealed that translation of CIP2A-BP is decreased by TGF-β. This micropeptide has an inhibitory role in the development of triple negative breast cancer possibly through suppression of PI3K/AKT/NF-κB signals and the subsequent down-regulation of matrix metalloproteinase (MMP)-2, MMP-9, and Snail levels [3].

Since this lncRNA has been dysregulated in different malignancies, it represents a potential target for therapeutic interventions. Thus, it is necessary to identify the functionally related molecules and pathways with this lncRNA in different tissues. In this manuscript, we have summarized the results of these studies and categorized them in three sections, i.e., cell line studies, animal model investigations, and investigations in clinical samples.

## 2. Cell Line Studies

LINC00665 has been shown to be up-regulated in acute myeloid leukemia (AML) cells parallel with the increase in expression of Dedicator Of Cytokinesis 1 (DOCK1) and decrease in expression of miR-4458. Knock-down of LINC00665 or DOCK1 has resulted in a significant decrease in proliferation, migration, and adhesion of these cells. On the other hand, suppression of miR-4458 has led to enhancement of these features and inhibition of apoptosis of AML cells. Mechanistically, LINC00665 could sponge miR-4458 and increase expression of DOCK1 through this route [4].

In breast cancer cells, LINC00665 has been found to act as a sponge for miR-379-5p, reducing the capacity of miR-379-5p to suppress expression of Lin-28 Homolog B (LIN28B). The impact of LINC00665 in induction of expression of LIN28B is associated with induction of progression of breast cancer and activation of epithelial–mesenchymal transition (EMT) program in these cells [5]. Another study in breast cancer cells has revealed attenuation of migration and invasion aptitude of these cells following LINC00665 silencing. Moreover, downregulation of this lncRNA has inhibited expressions of EMT-associated proteins in these cells [6]. LINC00665 can also influence progression of breast cancer via sponging miR-551b-5p [7]. LINC00665 silencing has also suppressed proliferation, migration, and invasive features of breast cancer cells, while it enhanced apoptosis. The effects of LINC00665 on these features of breast cancer cells are possibly exerted through sponging miR-3619-5p. Up-regulation of miR-3619-5p has been shown to be similar to LINC00665 silencing at cellular level. Expression of β-catenin has been reduced following LINC00665 silencing and miR-3619-5p up-regulation, supporting the importance of LINC00665/miR-3619-5p/β-catenin axis in the progression of breast cancer [8].

An in vitro experiment in cervical cancer cells has revealed that short hairpin RNA (shRNA)-mediated knock down of LINC00665 can lead to reduction of cell viability, up-regulation of E-cadherin level, down-regulation of N-cadherin, Vimentin and CTNNB1 levels, and suppression of migration and invasiveness of HeLa cells. The impact of LINC00665 on enhancement of EMT is possibly mediated via activation of WNT-CTNNB1/β-catenin signals [9].

A microarray-based assay in gemcitabine resistant cholangiocarcinoma cell lines has led to identification of LINC00665 among the top 10 over-expressed. Knock down of this lncRNA in chemoresistant cells has led to enhancement of gemcitabine effects, whereas up-regulation of LINC00665 has augmented gemcitabine resistance in chemosensitive cholangiocarcinoma cells. Chemoresistant cholangiocarcinoma cells have exhibited higher EMT and stemness features, and LINC00665 knock-down has inhibited sphere formation, migratory potential, invasiveness, and levels of EMT and stemness marker proteins. While Wnt/β-Catenin signals have been activated in chemoresistant cholangiocarcinoma cells, LINC00665 silencing has inhibited activity of this pathway. Mechanistically, LINC00665 can regulate expression of the nuclear transcriptional regulator of this pathway, i.e., BCL9L through sponging miR-424-5p. BCL9L down-regulation or miR-424-5p up-regulation has reduced resistance to gemcitabine, and decreased EMT, stemness, and Wnt/β-Catenin activity in chemoresistant cholangiocarcinoma cells [9].

In vitro experiments in colorectal cancer cells have shown the sponging role of LINC00665 on miR-9-5p and subsequent regulation of ATF1 as an important mechanism of carcinogenesis [10]. Moreover, LINC00665-mediated up-regulation of CTNNB1 has been demonstrated to result in activation of Wnt/β-catenin signals in colorectal cancer cells [11]. Finally, LINC00665 could stimulate proliferation and impede apoptosis of colorectal cancer cells through regulation of miR-126-5p expression [12].

Figure 1. Oncogenic role of LINC00665 in prostate cancer, colorectal cancer, breast cancer, and osteosarcoma. Detailed information about the assays is shown in Table 1.

In glioma cells, in vitro experiments have indicated down-regulation of TAF15 and LINC00665. Forced up-regulation of TAF15 has increased stability of LINC00665, suppressing malignant features in these cells. Moreover, MTF1 and YY2 transcription factor have been shown to be over-expressed in glioma cells, and their silencing has suppressed malignant behaviors of these cells. Up-regulation of LINC00665 could decrease stability of MTF1 and YY2 transcripts through interplay with STAU1, and STAU1 silencing has reversed LINC00665-mediated down-regulation of MTF1 and YY2. Finally, MTF1 or YY2 silencing has reduced expression of GTSE1 oncogene in glioma. Taken together, the TAF15/LINC00665/MTF1(YY2)/GTSE1 axis has been acknowledged as an important axis in the modulation of the malignant features of glioma cells [18]. Contrary to this study, another study has shown an oncogenic role for this lncRNA in glioma. This study has indicated that LINC00665 sponges miR-34a-5p to influence expression of AGTR1 [19].

In hepatocellular carcinoma, LINC00665 has been found to sponge miR-214-3p and increase expression of MAPK1, thus accelerating cell proliferation and Warburg effect [20]. Moreover, it can regulate the viability, apoptotic pathways, and autophagy of these cells through modulation of miR-186-5p/MAP4K3 [21] and Double-Stranded RNA–Activated Protein Kinase/NF-κB [22] axes. Another study in hepatocellular carcinoma cells has shown that LINC00665 silencing could decrease proliferation, migration, and invasion, whereas up-regulation the CIP2A-BP micropeptide enhances these features [36].

Figure 2. Role of LINC00665 in hepatocellular carcinoma, glioma, melanoma, gastric cancer, and lung cancer. Detailed information about the assays is shown in Table 1.

## 3. Animal Studies

Experiment in xengraft models of breast, colorectal, gastric, liver, lung, and prostate cancers as well as cholangiocarcinoma, endometrial carcinoma, and melanoma have confirmed that up-regulation of LINC00665 increases tumor burden, while its silencing decreases tumor weight (Table 2). Thus, these studies consistently point to the oncogenic effects of LINC00665.

## 4. Studies in Clinical Samples

LINC00665 has been shown to be over-expressed in breast cancer tissue samples in association with poor prognosis of breast cancer patients [6]. Similarly, up-regulation of LINC00665 in breast cancer samples has been correlated with tumor size and TNM stages in another cohort of breast cancer patients [8]. Over-expression of LINC00665 has also been associated with poor prognosis and resistance of cholangiocarcinoma patients to chemotherapy [9]. In liver cancer, expression of LINC00665 has also been elevated, which noticeably designated poor prognosis. Moreover, up-regulation of LINC00665 has been associated with further development of the tumors, which has been closely correlated with clinical diagnosis. Accuracy of LINC00665 levels in prediction of overall survival has also been verified by ROC curve analyses [36].

The association between over-expression of LINC00665 and poor survival of patients has been verified in breast cancer [6], cholangiocarcinoma [9], gastric cancer [15], glioma [19], hepatocellular carcinoma [20], lung cancer [24], osteosarcoma [30], ovarian cancer [33], and prostate cancer [34] (Table 3).

## 5. Conclusions

LINC00665 has been regarded as an oncogenic lncRNA in diverse tissues. In fact, apart from a single study on glioma [18], other studies have consistently demonstrated that LINC00665 facilitates proliferation and malignant behaviors of cancer cells. It is not clear whether LINC00665 has a tissue-dependent function in the carcinogenesis or if the finding on glioma is an exception to the oncogenic role of this lncRNA. The main way of contribution of this lncRNA in the carcinogenesis is its sponging impact on tumor suppressor miRNAs. miR-4458, miR-379-5p, miR-551b-5p, miR-3619-5p, miR-424-5p, miR-9-5p, miR-214-3p, miR-126-5p, miR-149-3p, miR-379-5p, miR-665, miR-34a-5p, miR-186-5p, miR-138-5p, miR-181c-5p, miR-98, miR-195-5p, miR-224-5p, miR-3619, miR-708, miR-101, miR-1224-5p, miR-34a-5p, and miR-142-5p are among the miRNAs that are sponged by LINC00665. Through modulating expression of these miRNAs, it can enhance expression of a number of oncogenes. Moreover, LINC00665 can influence activity of Wnt/β-Catenin, TGF-β, MAPK1, NF-κB, ERK and PI3K/AKT signaling. This lncRNA can also enhance tumor metastasis through activation of EMT program.

LINC00665 can also affect response to a number of anticancer drugs including gemcitabine, apatinib, and gefitinib. Thus, therapies targeted against LINC00665 are expected to decrease tumor volume, reduce malignant features, and improve the response of cancer cells to both chemotherapeutic drugs and targeted therapies. Although the results of investigations in animal models of diverse kinds of cancers have been promising, these results have not been validated in clinical settings due to several obstructions, particularly regarding safety and bioavailability issues. Moreover, LINC00665 has interactions with a wide range of biomolecules including miRNAs. Identification of other LINC00665 targets, particularly possible tissue-specific targets, is a prerequisite for the design of more specific therapeutic modalities.

An important note about this lncRNA is its ability in the production of a functional micropeptide. The encoded micropeptide by this lncRNA can promote progression of breast cancer and hepatocellular carcinoma. Future studies should assess the impact of this micropeptide in other types of cancers and find the possible functional relationship between lncRNA and micropeptide. Notably, expression of this micropepetide is regulated by TGF-β [3].

Finally, in spite of adequate data about the prognostic impact of LINC00665, there is no data about its role in diagnostic approaches. Thus, future studies should evaluate whether LINC00665 levels can separate patients with malignancies from normal subjects.

In conclusion, LINC00665 is an lncRNA that affects several aspects of carcinogenesis, particularly in response to both chemotherapeutic agents and small molecules that are used in targeted cancer therapy. Therefore, this lncRNA is a candidate for designing novel drugs for the treatment of cancer.

## Figures and Tables

**Figure 1 cells-11-03575-f001:**
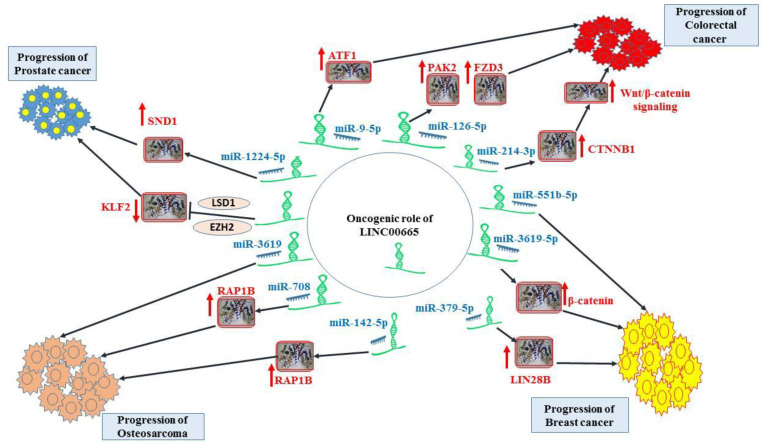
Demonstrates the oncogenic role of LINC00665 in prostate cancer, colorectal cancer, breast cancer and osteosarcoma.

**Figure 2 cells-11-03575-f002:**
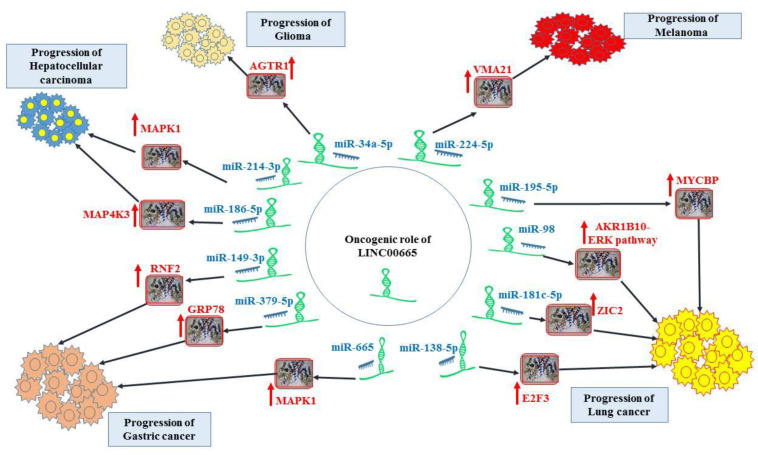
Shows the role of LINC00665 in liver cancer, glioma, melanoma, gastric cancer, and lung cancer.

**Table 1 cells-11-03575-t001:** LINC00665 expression in cell lines (∆: knock-down or deletion, CRC: colorectal cancer, DDP: cisplatin, AML: acute myeloid leukemia).

Tumor Type	Interactions	Cell Line	Function	Reference
**Acute myeloid leukemia**	miR-4458/DOCK1 axis	KG1, U937, NB4 and HL60 and HS-5	∆ LINC00665: ↓ proliferation, migration and adhesion and ↑ apoptosis	[4]
**Breast cancer**	miR-379-5p/LIN28B axis	MCF10A, 293FT, MCF7, BT474, BT549, MDA-MB-231, MDA-MB-468, and T47D	∆ LINC00665: ↓ proliferation, migration, and invasion ↑↑ LINC00665: ↑ proliferation, migration, and invasion, EMT process	[5]
	_	MCF-10A, MCF-7, MDA-MB-231, ZR-75-30, and MDA-MB-415	∆ LINC00665: ↓ migration, invasion and EMT process	[6]
	miR-551b-5p	MCF10A and HCC-1937, MDA-MB-231, and MCF-7	∆ LINC00665: ↓ cell growth and ↑ apoptosis	[7]
	miR-3619-5p/β-catenin axis	MDA-MB-231 and MCF-7 and MCF-10A	∆ LINC00665: ↓ cell proliferation, migration, and invasion	[8]
**Cervical cancer**	WNT-CTNNB1/β-catenin signaling pathway	HeLa and HEK293T cells	∆ LINC00665: ↓ cell viability, migration, invasion and EMT process	[9]
**Cholangiocarcinoma**	miR-424-5p/ BCL9L axis, Wnt/β-Catenin signaling	HuCCT1, HuH28, SNU-1196, SNU-1079, SNU-308, SNU-245, SNU-478 and SNU-869	∆ LINC00665: ↓ sphere formation, migration, invasion, EMT process, gemcitabine resistance, and stemness ↑↑ LINC00665: ↑ gemcitabine resistance	[9]
**Colorectal cancer**	miR-9-5p/ATF1 axis	DLD-1, SW480, KM12, SW116, SW620 and NCM460	∆ LINC00665: ↓ cell proliferation, migration, invasion and ↑ apoptosis ↑↑ LINC00665: ↑ cell proliferation, migration, invasion and ↓ apoptosis	[10]
	miR-214-3p/CTNNB1 axis, U2AF2, Wnt/β-catenin signaling pathway	NCM460, SW620, LoVo, HCT-116, SW480	∆ LINC00665: ↓ cell growth, migration and invasion, and ↑ apoptosis LINC00665 was found to up-regulate expression of CTNNB1.	[11]
	miR-126-5p/ PAK2 and FZD3 axis	DLD1, RKO, HCT116, LOVO, SW480 and NCM460	∆ LINC00665: ↓ proliferation and ↑ apoptosis	[12]
**Endometrial carcinoma**	HMGA1	RL-95-2, Ishikawa, HEC-1B, KLE and HHUA	∆ LINC00665: ↓ viability, migration and invasion, ↑ apoptosis and G1 phase arrest	[13]
**Gastric cancer**	Wnt signaling	MKN28, BGC-823, SGC-7901, AGS, HGC-27 and GES-1	∆ LINC00665: ↓ proliferation, migration, invasion, and ↑ apoptosis and cell cycle arrest	[14]
	miR-149-3p/RNF2 axis	AGS, SGC-7901, HGC27, MGC-803, MKN-45, BGC-823 and GES-1	∆ LINC00665: ↓ cell viability and invasion	[15]
	miR-379-5p/ GRP78 axis	GES-1, SGC-7901, AGS, HST2	∆ LINC00665: ↓ DDP-resistant GC cell proliferation, Endoplasmic reticulum (ER) stress, and ↑ apoptosis	[16]
	TGF-β signal pathway	AGS, MKN45, HGC27, and SGC7901, MKN28 and GES	∆ LINC00665: ↓ proliferation, invasion, metastasis, EMT process, ↑ cell apoptosis and G0/G1 phase arrest	[17]
	miR-665/MAPK1 axis	Apatinib-resistant gastric cancer cells	Paeonol treatment: ↓ LINC00665 levels, thus ↓ proliferation, migratory potential, invasive aptitude and glycolysis, and ↑ apoptosis of Apatinib-resistant gastric cancer cells	[16]
**Glioma**	TAF15, STAU1, MTF1 and YY2	U251, U87 glioma cells and HEK293T	LINC00665 Stabilized by TAF15 was found to destabilize MTF1 and YY2 transcripts via interplay with STAU1.	[18]
	miR-34a-5p/AGTR1 axis	U87 MG, LN229, A172, U373 MG, U251, NHA and 293T cells	↑↑ LINC00665: ↑ proliferation and invasion	[19]
**Hepatocellular carcinoma**	miR-214-3p/MAPK1 axis	SNU-387, SNU-423, SNU-449, and SNU-398	↑↑ LINC00665: ↑ cell viability migration, invasion and aerobic glycolysis	[20]
	miR-186-5p/MAP4K3 axis	Huh-7, HepG2, HCCLM6, MHCC-97H, Hep3B and HL-7702	∆ LINC00665: ↓ viability, ↑apoptosis and autophagy	[21]
	PKR, NF-κB signaling	EK-293T, Huh-7 and HepG2, and SK-Hep1, pWPXL-LINC00665	LINC00665 was found to stabilize PKR by blocking its ubiquitination and involvement in the activation of NF-κB signaling.	[22]
**Lung cancer**	miR-138-5p/E2F3 axis	A549, H520, H1299, SPC-A-1, SK-MES-1 and NHBE	∆ LINC00665: ↓ proliferation and invasion	[23]
	miR-181c-5p/ZIC2 axis	SK-LU-1 and Calu-3	∆ LINC00665: ↓ cell viability, clone formation, invasion and tumorigenesis	[24]
	YB-1-ANGPT4/ANGPTL3/VEGFA axis	HUVECs, A549 and H1299	Linc00665 was found to interact with YB-1 and induce angiogenesis in lung adenocarcinoma by activating YB-1-ANGPT4/ANGPTL3/VEGFA axis.	[25]
	miR-98/AKR1B10 axis and AKR1B10-ERK signaling pathway	A549, H1299, H1650, H520, SPCA-1, and SK-MES-1, 16HBE and HEK-293T	∆ LINC00665: ↓ proliferation, migration, and invasion	[26]
	miR-195-5p/MYCBP axis	HBE, A549, H1299, H1975, PC9, and SPCA-1	∆ LINC00665: ↓ proliferation, cell migration, invasion, and ↑ apoptosis	[27]
	EZH2 and PI3K/AKT pathway	PC9 and PC9/GR	∆ LINC00665: ↓ proliferation, ↑ apoptosis and gefitinib sensitivity	[28]
	EZH2 CDKN1C	16HBE, PC9, SPC-A1, H1975, H1299, and A549	∆ LINC00665: ↓ proliferation, migration ↑ apoptosis, G0/G1 phase arrest, and drug sensitivity of cells to DDP	[27]
**Melanoma**	miR-224-5p/VMA21 axis	A375, M21, A2058, A-875 and HEMa-LP	∆ LINC00665: ↓ proliferation and migration	[29]
**Osteosarcoma**	miR-3619	143B, U2OS, MG63 and Saos-2, hFOB1.19 and 293T cells	∆ LINC00665: ↓ viability, invasion, and migration	[30]
	miR-708 and miR-142-5p, RAP1B	MG63, U2OS, 143B and Saos-2 and hFOB	∆ LINC00665: ↓ proliferation, migration, and invasion	[31]
**Ovarian cancer**	miR-34a-5p/E2F3 axis	A2780, OVCAR3, CAOV3, SKOV3 and IOSE80	∆ LINC00665: ↓ proliferation, migration, and invasion	[32]
**Prostate cancer**	miR-1224-5p/SND1 axis	LNCaP, PC-3, DU-145, 22RV1 and RWPE-1	∆ LINC00665: ↓ growth and metastasis	[33]
	KLF2, EZH2 and LSD1	PC-3, DU-145, 22RV1, LNCaP and WPMY-1	∆ LINC00665: ↓ proliferation and migration LINC00665 was found to inhibit KLF2 expression by binding to EZH2 and LSD1.	[34]
**T cell acute lymphoblastic leukemia**	miR-101 and PI3K/Akt pathway	T-ALL cells	∆ LINC00665: ↓ viability, migration and invasion	[35]

↑ upregulation; ↓ Downregulation; ↑↑ significantly higher.

**Table 2 cells-11-03575-t002:** LINC00665 function in the carcinogenesis based on studies in animal models (∆: knock-down or deletion).

Tumor Type	Animal Models	Results	Reference
**Breast cancer**	5-week-old female SCID mice	∆ LINC00665: ↓ tumor volume ↑↑ LINC00665: ↑ tumor volume	[5]
	4-week-old BALB/c nude mice	∆ LINC00665: ↓ tumor growth	[7]
**Cholangiocarcinoma**	nude mice	∆ LINC00665: ↓Tumor growth, and tumor weight	[9]
**Colorectal cancer**	Female nude mice	∆ LINC00665: ↓ tumor growth, tumor volumes and tumor weights	[11]
**Endometrial carcinoma**	12 8-week-old female mice	∆ LINC00665: ↓ tumor growth and tumor volume	[13]
**Gastric cancer**	6-week-old male nude mice	∆ LINC00665: ↓ tumor weights and tumor development	[14]
	6-week-old BALB/c nude mice	∆ LINC00665: ↓ tumor growth	[17]
**Glioma**	4-week-old nude mice	↑↑ LINC00665: ↑ tumor growth and tumor weights	[19]
**Hepatocellular carcinoma**	6-week-old male BALB/c nude mice	∆ LINC00665: ↓ tumor volumes and weights	[20]
	female BALB/c nude mice	∆ LINC00665: ↓ tumor growth and tumor volume	[21]
**Lung cancer**	BALB/c nude mice	∆ LINC00665: ↓ tumor volumes, tumor weights and proliferation	[23]
	6–8-week-old BALB/c nude male mice	∆ LINC00665: ↓ tumor formation	[24]
	4-week-old female BALB/c athymic nude mice	∆ LINC00665: ↓ tumor size, tumor growth, tumor weight and metastasis	[26]
	male BALB/c nude mice	∆ LINC00665: ↓ tumor growth and metastasis	[27]
	5-week-old male athymic BALB/c nude mice	∆ LINC00665: ↓ tumor growth and ↑ gefitinib sensitivity	[28]
	4–5-week-old male BALB/c nude mice	∆ LINC00665: ↓ tumor size and tumor weight	[27]
**Melanoma**	6-week-old male BALB/C nude mice	∆ LINC00665: ↓ tumor volumes and weights	[29]
**Prostate cancer**	4-week-old female Balb/c nude mice	∆ LINC00665: ↓ tumor volumes and tumor weights	[33]
	8-week-old male nude mice	∆ LINC00665: ↓ tumor growth and tumor weights	[34]

↑ upregulation; ↓ Downregulation; ↑↑ significantly higher.

**Table 3 cells-11-03575-t003:** Dysregulation of LINC00665 in clinical samples (ANCTs: adjacent non-cancerous tissues, OS: overall survival, DFS: disease-free survival, BC: Breast cancer, pCR: pathological complete response, GC: Gastric cancer, PFS: progression-free survival, TNM: tumor node metastasis).

Tumor Type	Samples	Expression (Tumor vs. Normal)	Kaplan—Meier Analysis (Impact of LINC00665 Up-Regulation)	Univariate/ Multivariate Cox Regression	Association of LINC00665 Expression with Clinicopathologic Characteristics	Reference
**Acute myeloid leukemia**	36 patients and 36 healthy controls	Up-regulated	_	_	_	[4]
**Breast cancer**	TCGA database	Up-regulated	_	_	_	[5]
	GEPIA database60 pairs of tumors and ANCTs	Up-regulated	Shorter OS and DFS	_	tumor stage and tumor metastasis	[6]
	36 pairs of tumors and ANCTs	Up-regulated	Shorter OS	LINC00665 was found to be a possible biomarker to predict OS of BC patients.	TNM stage and lymph node metastasis	[7]
	SHPD002 study (102 advanced breast cancer patients)	Up-regulated	_	Linc00665 expression was found to be an independent predictor of pCR, especially in HR-positive/HER2-negative subtype patients.	lymph node metastasis	[37]
	106 pairs of tumors and ANCTs	Up-regulated	_	_	tumor size and tumor, node, and metastasis stages	[8]
**Cholangiocarcinoma**	100 pairs of tumors and ANCTs	Up-regulated	Shorter OS and recurrence-free survival time	_	higher TNM stage, lymph node involvement, and distant metastasis	[9]
**Colorectal cancer**	46 pairs of tumors and ANCTs	Up-regulated	_	_	local lymph node metastasis and poor differentiation	[10]
	67 pairs of tumors and ANCTs	Up-regulated	_	_	_	[12]
**Endometrial carcinoma**	10 pairs of tumors and ANCTs	Up-regulated	_	_	_	[13]
**Gastric cancer**	49 pairs of tumors and ANCTs	Up-regulated	Shorter OS	_	TNM stage, histological grade, and poor prognosis of GC patients	[15]
	GEO and TCGA databases	Up-regulated	Shorter OS and DFS	LINC00665 was found to be an independent prognostic biomarker in GC patients.	tumor depth, lymph node metastasis, and TNM stage	[17]
	GEPIA database, and GEO datasets (GSE109476 and GSE93415	Up-regulated	_	_	_	[16]
**Glioma**	48 pairs of tumors and ANCTs TCGA database	Up-regulated	Shorter OS	_	_	[19]
**Hepatocellular carcinoma**	50 pairs of tumors and ANCTs	Up-regulated	Shorter OS	_	_	[20]
	76 pairs of tumors and ANCTs	Up-regulated	Shorter OS	_	tumor size and Edmondson grade	[21]
	50 pairs of tumors and ANCTs GSE77314	Up-regulated	_	_	_	[22]
	TCGA, GEPIA and GEO databases 39 pairs of tumors and ANCTs	Up-regulated	Shorter OS	_	gender, histological grade, stage, and vascular invasion	[38]
**Lung cancer**	37 pairs of tumors and ANCTs	Up-regulated	_	_	TNM stage	[23]
	GEPIA and starBase databases 84 pairs of tumors and ANCTs	Up-regulated	Shorter OS	_	_	[24]
	60 pairs of tumors and ANCTs	Up-regulated	_	_	differentiation, tumor size, lymph node metastasis, TNM stage, and lymphovascular invasion	[25]
	80 pairs of tumors and ANCTs GEO database (GSE27262)	Up-regulated	Shorter OS and recurrence-free survival time	High levels of linc00665, positive lymph node metastasis, high TNM stage, were found to be independent prognostic factors for predicting poor recurrence-free survival in LUAD patients.	larger tumor size, advanced TNM stage, and lymph node metastasis	[26]
	TCGA database 52 pairs of tumors and ANCTs	Up-regulated	Shorter OS	_	poor prognosis and advanced T stage	[27]
	GEO database (GSE18842, GSE19188, and GSE33532)	Up-regulated	_	_	_	[39]
	20 patients	Up-regulated gefitinib-resistance	_	_	_	[28]
	TCGA database 60 pairs of tumors and ANCTs	Up-regulated	Shorter OS and PFS	_	advanced TNM stage, lymph node metastasis, and tumor size	[27]
**Osteosarcoma**	33 pairs of tumors and ANCTs	Up-regulated	Shorter OS	_	_	[30]
	42 pairs of tumors and ANCTs	Up-regulated	Shorter OS	_	larger tumor size and later clinical stages	[31]
**Ovarian cancer**	56 pairs of tumors and ANCTs TCGA database	Up-regulated	Shorter OS and PFS		tumor size, FIGO stage, and lymph node metastasis	[32]
	GEO database (GSE5438, GSE40595, GSE38666 and GSE26712)	Up-regulated	Shorter OS	_	_	[40]
**Prostate cancer**	41 pairs of tumors and ANCTs	Up-regulated	Shorter OS	_	_	[33]
	50 pairs of tumors and ANCTs	Up-regulated	Shorter OS	_	higher T stage and lymph node metastasis	[34]

## Data Availability

The analyzed data sets generated during the study are available from the corresponding author on reasonable request.

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
