# Peer review of "A Concise Review on Dysregulation of LINC00665 in Cancers"

_cells, 2022, doi:10.3390/cells11223575_

Round 1
Reviewer 1 Report
In this manuscript, the authors provide a detailed discussion on the roles of lncRNA LINC00665 in diverse cancers. The authors summarized the functionally related molecules and pathways that LINC00665 regulated in different malignancies, which can help to better understand the oncogenic role of LINC00665 in cancer progression. The manuscript is well written and the illustration is presented in a good quality. This will provide interesting information for the reader of the journal. However, there are still some grammatical and syntax errors in the article. So I think the manuscript can be accepted after grammar and language check.
Author Response
In this manuscript, the authors provide a detailed discussion on the roles of lncRNA LINC00665 in diverse cancers. The authors summarized the functionally related molecules and pathways that LINC00665 regulated in different malignancies, which can help to better understand the oncogenic role of LINC00665 in cancer progression. The manuscript is well written and the illustration is presented in a good quality. This will provide interesting information for the reader of the journal. However, there are still some grammatical and syntax errors in the article. So I think the manuscript can be accepted after grammar and language check.
Response: We edited the language of the manuscript.
Reviewer 2 Report
In the present review Ghaforui-Fard and co-workers summarise the results of several works on a gene for long intergenic non-protein coding RNA 665 that is located on chromosome 19. This is an extremely interesting topic for many scientist. However, I do not recommend this manuscript in its present format for publication in the Cells. The major reasons for this recommendation are as fellows:
The review is presented by three huge tables and two schemes specially. I do not find the schemes necessary. They do not bring any benefit.
The review contains a lot of abbreviations without explanation. Thus, reading of the review can be complicated for non-specialists.
In a review it is better to use Conclusion than Discussion.
Last but not least, the manuscript strikingly resembles the review already published in Cells by Zhong et al. (2022, 11, 1540). This is why, I recommend the authors to rewrite their review with focus on the fact that LINC00665 as well as other lncRNA can be translated and their peptid products can play important role in carcinogenesis.
Author Response
In the present review Ghaforui-Fard and co-workers summarise the results of several works on a gene for long intergenic non-protein coding RNA 665 that is located on chromosome 19. This is an extremely interesting topic for many scientist. However, I do not recommend this manuscript in its present format for publication in the Cells. The major reasons for this recommendation are as fellows:
The review is presented by three huge tables and two schemes specially. I do not find the schemes necessary. They do not bring any benefit.
Response: These figures represent some aspects of LINC00665 functions.
The review contains a lot of abbreviations without explanation. Thus, reading of the review can be complicated for non-specialists.
Response: We described the abbreviations.
In a review it is better to use Conclusion than Discussion.
Response: We used Conclusion instead of discussion.
Last but not least, the manuscript strikingly resembles the review already published in Cells by Zhong et al. (2022, 11, 1540). This is why, I recommend the authors to rewrite their review with focus on the fact that LINC00665 as well as other lncRNA can be translated and their peptid products can play important role in carcinogenesis.
Response: There are only two papers showing the impact of this peptide in the carcinogenesis. We discussed these papers.
Round 2
Reviewer 2 Report
The corrected review seems to be in order apart from its resemblance to a review by Zhong et al. appeared in Cells 2022, 11, 1540.